# The Influence of Vaginal Native Tissues Pelvic Floor Reconstructive Surgery in Patients with Symptomatic Pelvic Organ Prolapse on Preexisting Storage Lower Urinary Tract Symptoms (LUTS)

**DOI:** 10.3390/jcm9030829

**Published:** 2020-03-18

**Authors:** Ewa Rechberger, Katarzyna Skorupska, Tomasz Rechberger, Małgorzata Wojtaś, Paweł Miotła, Beata Kulik-Rechberger, Andrzej Wróbel

**Affiliations:** 1Second Department of Gynecology, Medical University of Lublin, Jaczewskiego 8, 20-954 Lublin, Poland; ewarechberger92@gmail.com (E.R.); rechbergt@yahoo.com (T.R.); malgorzata.woytas@gmail.com (M.W.); pmiotla@wp.pl (P.M.); wrobelandrzej@yahoo.com (A.W.); 2Department of Paediatric Propedeutics, Medical University of Lublin, Gębali 6, 20-093 Lublin, Poland; brechberger@interia.pl

**Keywords:** vaginal native tissue repair, pelvic organ prolapse, lower urinary tract symptoms

## Abstract

The aim of this study was to assess the effectiveness of vaginal native tissue repair (VNTR) on preexisting Lower Urinary Tract Symptoms (LUTS) in women with symptomatic pelvic organ prolapse (POP). Two hundred patients who underwent VNTR for symptomatic POP from January 2018 to February 2019 were followed up for 6 months. All patients underwent VNTR, but in the posterior compartment, the rectovaginal fascia was reconnected to the uterosacral ligaments and additionally sutured to the iliococcygeus fascia and muscle. To assess the severity and change in storage phase LUTS before and after surgery, all participants were asked to complete 3 questionnaires: the International Consultation on Incontinence Questionnaire- Short Form (ICIQ-SF), Urogenital Distress Inventory-6 (UDI-6), and Incontinence Impact Questionnaire-7 (IIQ-7). The data were assessed with Statistica package version 12.0, using Kalmogorow–Smirnoff, W Shapiro–Wilk tests. Furthermore, one-way analysis of variance was applied with post-hoc Tukey test. The study results indicate that the majority of patients with advanced POP suffered from various LUTS. Among storage symptoms, the occurrence of urinary incontinence (UI) and urgency decreased significantly after surgery. Moreover, ICIQ-SF, UDI-6, and IIQ-7 results showed statistically significant improvement in the impact of UI on the quality of life (QoL) in the vast majority of patients after surgery. VNTR is an effective way to treat not only anatomical, but also functional problems in such patients.

## 1. Introduction

Pelvic organ prolapse (POP) and accompanying functional disorders are significant epidemiological, medical, and social problems. Contemporary reports estimate the prevalence of POP to be between 3% and 6% [1]. POP is a common phenomenon often requiring surgical correction due to the bothersome symptoms reported by patients [2,3]. A correlation has been noted between the occurrence and severity of lower urinary tract symptoms (LUTS) and the clinical advancement of POP [2]. Herein, it was found that patients with advanced prolapse who underwent reconstructive surgery are also at risk of developing undesired postoperative LUTS if they are over the age of 66, are burdened with additional neurological factors and postvoid residual (PVR) ≥ 200 mL [4].

The etiology of the prolapse is multifactorial. It is generally associated with the pressure exerted by internal organs (urinary bladder, uterus, intestines) on the pelvic floor, whose biomechanical strength is impaired due to congenital genetic defects, previous surgical procedures, or neurodegenerative changes resulting from perinatal damage or neurological disorders [5]. In the case of advanced prolapse, the only really effective way of treatment is reconstructive pelvic surgery. Although the majority of women accept short-term use of pessaries for the temporary treatment of symptomatic POPs, this method is not accepted in long-term use due to lack of permanent effectiveness [6].

Clinicians should remember that even vaginal native tissue repair (VNTR) may be associated with higher rates of recurrence when compared to synthetic mesh repairs, but POP surgery with mesh has definitely higher postoperative complication rates [7,8]. Occurrence of postoperative urinary incontinence (UI) is quite well discussed in the available literature, but much less is written about the occurrence of other LUTS either before or after pelvic reconstructive surgery [9].

The demographic trends clearly show the aging of highly developed societies with an obvious increase in the need for safe and effective reconstructive pelvic surgery. Nowadays, the changing surgical trends should be mainly focused on the safety and reversibility of potential complications, and not only on the minimalization of the risk of recurrence [10]. Therefore, currently, the first-line operations in the treatment of POP are again surgeries using the patient’s own tissues and delayed resorbable or nonresorbable sutures [11]. Even sacrocolpopexy with polypropylene mesh, which is currently considered as a gold standard for treatment of advanced apical prolapse, is not categorically better than VNTR with regard to functional outcomes [12,13]. Keeping the above in mind, the primary aim of the present study was to assess the influence of VNTR on the preexisting common LUTS in patients with symptomatic pelvic organ prolapse (POP-Q II-IV). Moreover, the influence of such surgery on the patients’ general quality of life (QoL) was assessed.

## 2. Materials and Methods

The study protocol was conducted in accordance with the European Communities Council Directive of 22 September 2010 (2010/63/EU) and Polish legislation acts, and was approved by the Local Ethics Committee (KE-0254/75). Before inclusion, all patients gave written informed consent for the participation in the study.

The main inclusion criterium was the diagnosis of symptomatic POP (mainly bulging) in the second or higher degree according to the Pelvic Organ Prolapse Quantification System (POP-Q) with or without coexisting overt or latent stress urinary incontinence (SUI) [14]. The exclusion criteria from the study were LUTSs resulting from pathologies unrelated to the POP (UTIs, stones, tumors, neurological diseases, urogenital atrophy). In addition, significant post void residual (more than 200 mL) was an exclusion criterium. The study involved 200 patients treated in the Single Gynecology Center from January 2018 to February 2019. Demographic data of the patients are presented in Table 1. During the physical examination, a cough and Valsav’a tests were performed after filling the bladder up to a volume of 200 mL to diagnose concomitant SUI. This test was repeated after repositioning of the vaginal walls with the use of the Kalmorgen double-sighted speculum. Patients with overt or occult SUI were informed that in case of persistent symptoms after VNTR, polypropylene Transobturator Tape (TOT) will be inserted in the future, in the outpatient department. The assessment of the degree of POP was based on the POP-Q scale [14].

During detailed medical history-taking, all eligible participants were asked before and after surgery for occurrence of common LUTS, using Patient Global Impression of Severity (PGI-S- with response options: 0- not present, 1- very mild, 2- mild, 3- moderate, 4- strong, 5- very strong) and Patient Global Impression of Improvement (PGI-I with response options ‘very much better’, ‘much better’, ‘a little better’, ‘no change’, ‘a little worse’, ‘much worse’, and ‘very much worse’) in order to assess how annoying for the patients these symptoms were and to assess change in the patient’s condition. PGI-S concerning LUTS was completed before, as well as six weeks and six months after surgery. PGI-I was completed 6 weeks and 6 months after surgery. Answers “0” and “1” were considered as clinically insignificant, “2” and “3”-moderate, while “4” and “5” were assessed as very annoying for the patients.

The participants of the study were placed within four groups depending on the urinary storage phase disorder [15]. The first group (55 women, 27.5% of all participants of the study) consisted of patients complaining of urgency. The second group (20 women, 10% of the studied population) showed concomitant SUI. The third group was made up of women who reported both symptoms of SUI and urgency (MUI)- 72 women (36% of the study population). In the fourth subgroup (53 women, 26.5% of the study participants) were patients who did not report clinically significant concomitant functional disorders of the LUT.

All participants were additionally asked to complete 3 questionnaires: the International Consultation on Incontinence Questionnaire- Short Form (ICIQ-SF), Urogenital Distress Inventory-6 (UDI-6), and Incontinence Impact Questionnaire-7 (IIQ-7). The last two assessed the impact of reconstructive surgery on preexisting incontinence [16,17,18]. The ICIQ-SF assesses the severity and impact of UI on the QoL [19]. It is widely regarded as the “gold standard” in the diagnosis of UI and commonly used in randomized trials. It should be mentioned that all questionnaires used in this study were validated in the Polish population, and, therefore, their reliability was confirmed [20].

After a detailed clinical assessment, all patients underwent POP repair using the VNTR technique. Briefly, a vaginal submucosal infiltration with diluted epinephrine solution (1/400.000) was performed. The surgical technique in case of anterior defect ≥III stage included an anterior median longitudinal colpotomy in order to reach the pubocervical fascia. The anterior repair was performed by placing two layers of 2–0 synthetic absorbable sutures at the pubocervical fascia. The surplus of the vaginal epithelium was removed and the vaginal mucosa was closed with continuous absorbable suture. In case of posterior defect ≥II stages, a vertical incision in the posterior vaginal mucosa was made and the rectovaginal fascia was identified and reconnected to the uterosacral ligaments at the top of the vagina and finally sutured to the iliococcygeus fascia and muscle inferiorly to the ischial spines [13]. The rectovaginal fascia was repaired with single absorbable sutures. Finally, reconstruction of the perineal body was performed. At the end of operation, vaginal skin closure was performed with delayed absorbable sutures.

In the perioperative period, each patient received 1 g of cefazolin iv, followed by metronidazole in the dose of 500 mg iv for 2 consecutive days. In addition, antithrombotic prophylaxis (Enoxaparinum 40 mg) was used perioperatively. Median postoperative hospital stay was 2 days. No severe intraoperative complications occurred. Six weeks and 6 months after the procedure, a detailed analysis including an interview and physical examination was conducted. The patients again filled in the questionnaires mentioned above. The pre- and postoperative assessment of study subjects and surgical procedures were performed by the same study team.

### Statistical Analysis

Statistical analysis was performed using STATISTICA 12.0 PL package. Kalmogorow–Smirnoff with Lilliefors modification and W Shapiro–Wilk tests were used in order to check compliance with normal distribution. Furthermore, one-way analysis of variance was used to ascertain the difference between variables, together with post-hoc Tuckey test. r-Pearson’s test was used to calculate correlation and the cluster analysis was used as a grouping test. P < 0.05 was considered as statistically significant.

## 3. Results

The POP-Q assessment revealed 58 (29%) women at stage II, 117 (58.5%) at stage III, and 25 (12.5%) at stage IV. The demographic data off all patients were presented in Table 1.

It should be stressed that majority of patients with advanced POP were also suffering from various LUTS. The evolution of preexisting LUTS following VNTR are presented in Table 2**.**

The analysis of the patients with urgency (Group 1) carried out 6 weeks and 6 months after the operation showed a significantly lower severity of urgency compared to the period before surgery. There were no statistically significant differences in urgency between 6 weeks and 6 months after surgery in the examined group of women. In addition, in both analyzed postoperative periods compared to the period before the surgery, we confirmed significantly less voiding episodes during the day and less frequent urination at night. There were no statistically significant differences in the frequency of voiding during the day and urination at night between 6 weeks and 6 months after surgery. Clinically, complete relief of urgency was obtained in 26 women (47%) after 6 weeks, and **in** 30 (55%) women 6 months after surgery.

The analysis carried out 6 weeks and 6 months after surgery in women with MUI showed a significant reduction in the number of voids during the day and less frequent urination at night compared to the preoperative period. There were no statistically significant differences in the frequency of micturition during the day and urination at night between 6 weeks and 6 months after surgery in the examined group of women. A statistically significant decrease was also shown in relation to the feeling of urgency six months after surgery. However, no such relationship was found by comparing the periods before and after 6 weeks after surgery and between 6 weeks and 6 months after surgery. Importantly, the severity of SUI significantly decreased when comparing the preoperative period with the period of 6 weeks after surgery. Clinically complete MUI resolution was obtained in 39 women (54.16%) 6 weeks post-surgery, and 35 women (48.61%) 6 months after surgery.

In the group of women without clinically significant functional complaints from the LUT, a statistically significant decrease in the severity of SUI was found between the state before surgery and 6 weeks afterwards. This is obviously due to the fact that in the pre-operative survey patients marked the answers “0” or “1”, which were considered clinically insignificant. However, the surgery proved to have also a beneficial effect on the severity of mild SUI and further alleviated the patient’s discomfort. There were no statistically significant differences in the frequency of voiding episodes during the day and urination at night between the studied periods.

Table 3 presents PGI-I results in study groups and Table 4 and Table 5 present the relationship between PGI-I and the severity of functional disorders 6 weeks and 6 months after surgery.

In the group of women with MUI, better PGI-I score 6 weeks after surgery is associated with a lower severity of LUTS, lower severity of UI and its effect on activity, interpersonal relationships, and feelings, and lower intensity of leaking. In women with SUI and those without LUTS, better PGI-I score 6 weeks after surgery coincides with lower intensity of leaking, while in the group with urgency with lower intensity of LUTS.

In the group of women with MUI and SUI, better PGI-I score 6 months after surgery is associated with lower severity of LUTS, lower severity of UI and its effect on activity, relationships, and feelings, and lower intensity of leaking. In women with urgency, better PGI-I score 6 months after surgery is associated with lower severity of LUTS, lower intensity of UI and its effect on activity, interpersonal relationships, and feelings, while in the group without UI with lower intensity of UI and its impact on activity, interpersonal relationships, and feelings, and less severe leaking.

In Table 6, results of UDI-6, IIQ-7, and the ICIQ-SF questionnaires are presented.

According to the UDI-6 and IIQ-7 results, patients with urgency had significantly lower severity of urinary function disorders in the analysis carried out 6 weeks and 6 months after surgery, compared to the preoperative period. No such relationship was found comparing the two postoperative periods.

In patients suffering additionally from SUI, a significantly lower severity of urinary functional disorders assessed by the UDI-6 questionnaire six months after surgery, compared to the period before surgery, was demonstrated. This relationship was not confirmed by assessing the 6-week period after surgery with the preoperative period and the two postoperative periods between each other. In the IIQ-7 results, there were no statistically significant differences between the period before surgery and 6 weeks or 6 months after surgery and between each period after surgery.

According to the UDI-6 questionnaire, women from the MUI group had significantly lower severity of urinary function disorders on comparing both postoperative periods with the period before surgery. In this group of patients, no statistically significant differences were found between the analyzed postoperative periods. IIQ-7 results showed significantly lower intensity of UI and its lower impact on activity, interpersonal relationships, and feelings in the analysis carried out 6 months after surgery, compared to the preoperative period. There were no statistically significant differences in this group between the 6 weeks and 6 months after surgery and between the pre-surgery and 6 weeks after surgery.

ICIQ-SF results in patients with urgency showed significant reduction in the severity of this symptom in both postoperative periods, compared to the period before surgery. There were no statistically significant differences in the severity of urge urinary incontinence between the compared periods after reconstruction surgery in this group. Patients from the SUI group demonstrated significant decrease in severity of UI within 6 months after surgery compared to the preoperative period. This relationship was not demonstrated by comparing the 6 weeks postoperative period with the pre-operative period and the two postoperative periods with each other. In the group of MUI, a significantly lower intensity of UI was noted both 6 weeks and 6 months after surgery compared to the period before surgery. In contrast, a comparison of both analyzed periods after reconstructive surgery did not show any statistically significant severity of UI.

## 4. Discussion

Keeping in mind the controversy regarding the use of meshes for POP surgery, by choosing first VNTR, we strictly followed the guidelines that synthetic meshes for treating prolapse should be used only in complex cases with recurrent prolapse [8]. In the present study, we were mainly focused on the assessment of the impact of VNTR, performed due to symptomatic POP, on the incidence and severity of various LUTS. In the last decades, much more attention was paid on QoL issues connected with the surgical treatment of pelvic floor disorders. That means that the definition of success and the methodology of final outcomes assessment have evolved and definitely are not based only on anatomic results, but also on functional outcome measures and patient-centered results as well. This is in line with the recently proposed evaluation of bladder health being defined as “a complete state of physical, mental, and social well-being related to bladder function, and not merely the absence of LUTS” [21]. Therefore, pelvic health should be defined not only as the lack of prolapse, but also proper functions—including urine and stool continence and sexual functions as well.

The obtained results showed, among others, a statistically significant decrease in the incidence of urgency in the early postoperative period (6 weeks) and the persistence of this effect 6 months after the surgery. A significant reduction was also observed in relation to the incidence and severity of urge and SUI after VNTR. In a recently published review, Peyronnet et al. [22] proposed patient-centered individualized-tailored treatment for overactive bladder (OAB), however, they do not even mentioned patients with symptomatic prolapse who, in majority of cases, will be cured from OAB symptoms by means of properly performed reconstructive surgery [23]. This was also the case in our study—where more than 50% of all affected individuals were cured from urgency after VNTR. The obtained results also proved that urinary outflow disturbances found in women with POP, in particular in the anterior vaginal wall, play an important role in the etiopathogenesis of female OAB, even though POP is not recognized as obvious pathology in the current definition of OAB. This finding is in line with the previous results that showed that POP reduction leads to improved urethral flow and withdrawal of detrusor overactivity symptoms [24]. This also reveals that POP is an independent risk factor for the emergence of OAB symptoms. It has been also shown previously that the higher POP-Q grade, the greater the likelihood of bladder outlet obstruction and increased PVR, but without a clear increase in other LUTS [25]. In a 4-year prospective study, progression from grade II to higher POP-Q grades was seen to be positively correlated with the severity of obstructive symptoms and increased constipation rate [26]. It has also been shown that patients with rectocele are more likely to experience defecation and constipation disturbances, while other LUTS do not show such correlation [27].

There is no doubt that patients with severe preoperative symptoms, muscular dysfunctions of the pelvic floor with neurological disorders, and a significant grade of prolapse (especially the anterior vaginal wall) are at higher risk for both pre- and postoperative LUTS [28]. It has been also indicated that the use of pessary is an effective and inexpensive diagnostic test allowing to predict in 20% of pre-operatively tested women, the occurrence of urgency, frequency, and UI after surgery [29].

In this study, we focus on patient-centered results, seeing them equal or even more significant than pure anatomical results in evaluating final surgical outcome. We found in our study group an overall high rate of efficacy for VNTR, with almost no intraoperative and postoperative complications. Therefore, in our opinion, VNTR procedures should not be regarded as a compromise. In fact, the present study strongly supports the efficacy and safety of the VNTR for the treatment of commonly occurring LUTS (especially urgency) in patients with advanced POP. In the context of our data and the FDA warnings, the use of the VNTR should be recommended in the majority of women with advanced POP. The definitive strength of our study is its prospective design and the fact that we followed strictly the International Continence Society/International Urogynecological Association (ICS/IUGA) guidelines for surgical outcomes using already validated tools, such as questionnaires for prolapse symptoms; urinary, defecatory, and sexual function; and patient satisfaction [30]. The main limitation of the study was the single setting and lack of control group, but, on the other hand, the single setting warrants that the applied surgical technique was the same for every patient, and, therefore, the final outcome, whether anatomical or functional, could be compared.

## 5. Conclusions

The majority of women with symptomatic POP are also suffering from various LUTS. VNTR is an effective way to treat not only anatomical, but also functional problems in such patients.

## Figures and Tables

**Table 1 jcm-09-00829-t001:** Patients demographic depending on concomitant dominant storage phase disorders of the lower urinary tract -based on the Lower Urinary Tract Symptoms (LUTS) survey.

Storage Phase Symptoms	*n*	PGI-S Mean ± SD	VD Mean ± SD	CC Mean ± SD	Age Mean ± SD	BMI Mean ± SD
Urgency	55	4.09 ± 1.43	2.35 ± 0.95	0.27 ± 0.47	58.69 ± 0.58	27.38 ± 3.81
SUI	20	3.05 ± 1.39	2.00 ± 1.18	0.25 ± 0.5	54.2 ± 10.43	26.10 ± 2.87
MUI	72	3.83 ± 1.34	2.59 ± 0.97	0.38 ± 0.65	60.71 ± 8,85	28.58 ± 6.94
NONE	53	0.45 ± 0.49	2.04 ± 1.00	0.17 ± 0.41	56.34 ± 10.86	27.80 ± 4.37

*n*- number of patients, CC- cesarean section, VD- vaginal deliveries, PGI-S- Patient Global Impression of Severity, SUI- stress urinary incontinence; MUI- mix urinary incontinence.

**Table 2 jcm-09-00829-t002:** The evolution of storage lower urinary tract symptoms (LUTS) after vaginal native tissue repair (VNTR) in studied groups.

	(T1) Before the Procedure	(T2) 6 Weeks After the Procedure	(T3) 6 Months After the Procedure	ANOVA	Post-Hoc
	PGI-S M ± SD	PGI-S M ± SD	PGI-S M ± SD		
Group 1- Urgency					
Urgency	2.09 ± 1.84	1.13 ± 1.49	0.91 ± 1.48	F _(2.108)_ = 14.28 *p* < 0.001	**T1 vs. T2: *p* < 0.001** **T1 vs. T3: *p* < 0.001** T2 vs. T3: NS
Increased Frequency of Urination	2.51 ± 1.51	1.89 ± 1.29	1.93 ± 1.32	F _(2.108)_ = 8.36 *p* < 0.001	**T1 vs. T2: *p* <0.001** **T1 vs. T3: *p* = 0.002** T2 vs. T3: NS
Nocturia	2.33 ± 1.65	1.75 ± 1.54	1.78 ± 1.63	F _(2.108)_ =3.95 *p* = 0.02	**T1 vs. T2: *p* = 0.04** **T1 vs. T3: *p* < 0.05** T2 vs. T3: NS
Group 2- SUI					
SUI	2.75 ± 1.48	1.85 ± 1.72	1.39 ± 1.76	F _(2.38)_ =3.86 *p* < 0.001	T1 vs. T2: NS **T1 vs. T3: *p* < 0.001** T2 vs. T3: NS
Increased Frequency of Urination	2.70 ± 1.66	2.10 ± 1.65	2.15 ± 1.57	F _(2.38)_ =1.35 *p* = NS	T1 vs. T2: NS T1 vs. T3: NS T2 vs. T3: NS
Nocturia	1.35 ± 1.23	1.30 ± 1.72	1.30 ± 1.53	F _(2,38)_ = 1.35 *p* = NS	T1 vs. T2: NS T1 vs. T3: NS T2 vs. T3: NS
Group 3- MUI					
Urgency	2.85 ± 1.58	2.47 ± 1.59	2.23 ± 1.79	F _(2.142)_ = 6.21 *p* = 0.003	T1 vs. T2: NS **T2 vs. T3: *p* = 0.001** T2 vs. T3: NS
SUI	3.04 ± 1.74	2.46 ± 1.8	2.5 ± 1.9	F _(2.142)_ = 6.47 *p* < 0.00	**T1 vs. T2: *p* = 0.003** T1 vs. T3: NS T2 vs. T3: NS
Increased Frequency of Urination	3.53 ± 1.42	3.13 ± 1.52	3.06 ± 1.63	F _(2.142)_ = 4.39 *p* = 0.01	**T1 vs. T2: *p* < 0.05** **T1 vs. T3: *p* = 0.02** T 2 vs. T 3: NS.
Nocturia	2.89 ± 1.50	2.19 ± 1.67	2.15 ± 1.86	F _(2.142)_ = 7.53 *p* < 0.001	**T1 vs. T2: *p* = 0.003** **T1 vs. T3: *p* = 0.002** T2 vs. T3: NS
Group 4- No Clinically Significant LUTS Symptoms					
Urgency	0.87 ± 1.21	0.88 ± 1.5	0.92 ± 1.55	F _(2.142)_ = 0.07 *p* = 0.93	T1 vs. T2: NS T1 vs. T3: NS T2 vs. T3: NS
SUI	1.13 ± 1.13	0.79 ± 1.34	0.73 ± 1.3	F _(2.104)_ = 3.88 *p* < 0.02	T1 vs. T2: NS **T1 vs. T3 *p* = 0.03** T2 vs. T3: NS
Increased Frequency of Urination	2.09 ± 1.55	2.06 ± 1.83	1.83 ± 1.66	F _(2.104)_ = 1.08 *p* = NS	T1 vs. T2: NS T1 vs. T3: NS T2 vs. T3: NS
Nocturia	1.70 ± 1.59	1.45 ± 1.60	1.30 ± 1.34	F _(2.104)_ = 1.98 *p* = NS	T1 vs. T2: NS T1 vs. T3: NS T2 vs. T3: NS

M- median; SD- standard deviation; SUI- stress urinary incontinence; MUI- mix urinary incontinence., T1- time point 1 before the VNTR procedure; T2- time point 2–6 weeks after the VNTR; T3- time point 3–6 months after the VNTR. Values marked in bold indicate clinically significant changes.

**Table 3 jcm-09-00829-t003:** Patient Global Impression of Improvement (PGI-I) results in study groups.

Time Point	PGI-I						
	Very Much Better	Much Better	Minimally Better	No Change	Minimally Worse	Much Worse	Very Much Worse
Group 1- Urgency							
(T2) 6 Weeks After the Procedure	11 (20%)	26 (47%)	8 (14.6%)	5 (9%)	4 (7.3%)	1 (1.8%)	0
(T3) 6 Months After the Procedure	13 (24%)	23 (42%)	7 (13%)	6 (11%)	5 (9%)	1 (1.8%)	0
Group 2- SUI							
(T2) 6 Weeks After the Procedure	2 (10%)	6 (30%)	3 (15%)	5 (25%)	3 (15%)	1 (5%)	0
(T3) 6 Months After the Procedure	3 (15%)	6 (30%)	4 (20%)	3 (15%)	3 (15%)	1 (5%)	0
Group 3- MUI							
(T2) 6 Weeks After the Procedure	11 (15.3%)	24 (33.3%)	17 (24%)	12 (17%)	7 (9.7%)	1 (1.4%)	0
(T3) 6 Months After the Procedure	10 (13.8%)	27 (37.5%)	15 (21%)	11 (15.3%)	8 (11.1%)	1 (1.4%)	0
Group 4- No Clinically Significant LUTS Symptoms							
(T2) 6 Weeks After the Procedure	0	0	6 (11.3%)	42 (79.2%)	4 (7.5%)	1 (1.8%)	0
(T3) 6 Months After the Procedure	0	0	6 (11.3%)	42 (79.2%)	4 (7.5%)	1 (1.8%)	0

**Table 4 jcm-09-00829-t004:** The relationship between Patient Global Impression of Improvement (PGI-I) and the severity of functional disorders 6 weeks after surgery.

Questionnaire	PGI-I After 6 Weeks							
	Urgency		SUI		MUI		No symptoms	
UDI-6	**0.33**	***p* = 0.015**	**0.31**	*p* = 0.178	0.27	*p* = 0.020	0.03	*p* = 0.821
IIQ-7	0.20	*p* = 0.152	0.43	*p* = 0.056	0.30	*p* = 0.010	0.21	*p* = 0.134
ICIQ-SF	0.22	*p* = 0.108	0.48	*p* = 0.034	0.35	*p* = 0.002	0.29	*p* = 0.035

SUI-stress urinary incontinence; MUI- mix urinary incontinence, UDI-6 - Urogenital Distress Inventory-6; IIQ-7- Incontinence Impact Questionnaire-7, ICIQ-SF- International Consultation on Incontinence Questionnaire- Short Form; Values marked in bold indicate clinically significant changes.

**Table 5 jcm-09-00829-t005:** The relationship between Patient Global Impression of Improvement (PGI-I) and the severity of functional disorders 6 months after surgery.

Questionnaire	PGI-I After 6 Months							
	Urgency		SUI		MUI		No Symptoms	
UDI-6	**0.3396**	*p* = 0.011	0.5165	*p* = 0.020	0.5267	*p* = 0.000	0.1893	*p* = 0.175
IIQ-7_	0.2979	*p* = 0.027	0.6401	*p* = 0.002	0.4142	*p* = 0.000	0.3425	*p* = 0.012
ICIQ-SF	0.0011	*p* = 0.994	0.6402	*p* = 0.002	0.4488	*p* = 0.000	0.272	*p* = 0.049

SUI- stress urinary incontinence; MUI- mix urinary incontinence, UDI-6- Urogenital Distress Inventory-6; IIQ-7- Incontinence Impact Questionnaire-7, ICIQ-SF- International Consultation on Incontinence Questionnaire- Short Form.

**Table 6 jcm-09-00829-t006:** Mean results of Urogenital Distress Inventory-6 (UDI-6), Incontinence Impact Questionnaire-7 (IIQ-7), and the International Consultation on Incontinence Questionnaire-Short Form (ICIQ-SF) among patients from the study groups.

Time Point		T1	T2	T3	Tests	
Questionnaire	Study Group	Mean Score before the Procedure ± SD	Mean Score 6 Weeks after the Procedure ± SD	Mean Score 6 Months after the Procedure ± SD	ANOVA	Post-Hoc
UDI-6	Urgency	35.96 ± 23.88	20.40 ± 16.49	19.90 ± 22.65	F _(2.108)_ = 18.66 *p* = 0.001	**T1 vs. T2: *p* < 0.001** **T1 vs. T3: *p* < 0.001** T2 vs. T3: NS
	SUI	42.77 ± 19.17	32.37 ± 20.60	25.70 ± 19.83	F _(2.38)_ = 7.30 *p* = 0.002	T1 vs. T2: NS **T1 vs. T3: *p* = 0.002** T2 vs. T3: NS.
	MUI	54.62 ± 25.84	45.33 ± 27.11	44.79 ± 29.06	F _(2.142)_ = 9.48 *p* = 0.001	**T1 vs. T2: *p* < 0.001** **T1 vs. T3: *p* < 0.001** T2 vs. T3: NS
IIQ-7	Urgency	28.22 ± 28.26	20.34 ± 23.41	20.34 ± 26.12	F _(2.108)_ = 4.83 *p* = 0.009	**T1 vs. T2: *p* = 0.022** **T1 vs. T3: *p* = 0.022** T2 vs. T3: NS
	SUI	30.71 ± 24.55	27.38 ± 22.49	19.76 ± 27.39	F _(2.38)_ = 2.71 *p* = NS	T1 vs. T2: NS T1 vs. T3: NS T2 vs. T3: NS
	MUI	49.20 ± 27.76	44.44 ± 28.04	42.79 ± 29.99	F _(2.142)_ = 3.87 *p* = 0.023	T1 vs. T2: NS **T1 vs. T3: *p* = 0.021** T2 vs. T3: NS
ICIQ-SF	Urgency	4.35 ± 3.96	3.15 ± 3.76	2.71 ± 3.62	F _(2.108)_ = 8.06 *p* < 0.001	**T1 vs. T2: *p* = 0.015** **T1 vs. T3: *p* < 0.001** T2 vs. T3: NS
	SUI	7.65 ± 3.90	6.05 ± 4.44	5.15 ± 4.98	F _(2.38)_ = 5.90 *p* = 0.006	T1 vs. T2: NS **T1 vs. T3: *p* = 0.005** T2 vs. T3: NS
	MUI	9.85 ± 5.44	8.63 ± 5.19	7.96 ± 5.43	F _(2.142)_ = 15.10 *p* < 0.001	**T1 vs. T2: *p* = 0.001** **T1 vs. T3: *p* < 0.001** T2 vs. T3: NS

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
