# Peer review of "The Influence of Vaginal Native Tissues Pelvic Floor Reconstructive Surgery in Patients with Symptomatic Pelvic Organ Prolapse on Preexisting Storage Lower Urinary Tract Symptoms (LUTS)"

_jcm, 2020, doi:10.3390/jcm9030829_

Round 1
Reviewer 1 Report
This is an interesting paper on the influence of vaginal native tissues pelvic floor reconstructive surgery in patients with symptomatic pelvic organ prolapse on preexisting storage Lower Urinary Tract Symptoms.
The study is well-designed with a good sample size. The outcome measures are carefully selected. The subject is useful as it evaluates meshless techniques for vaginal POP surgery, which are expected to become increasingly prevalent.
Its main defect is that it was carried out with an unacceptably short follow-up. Six months is simply not enough time for us to know whether the results of surgery will last. A study of this kind should continue to monitor patients for at least two years. If it is possible for you to extend your study, I would be happy to recommend it for publication at that point.
Secondly, you used PGI-S pre- and post-operatively to monitor the effect of surgery on LUTS, but this is an inappropriate and you should have used PGI-I, which was designed to assess change in the patient’s condition, whereas the PGI-S only assesses severity (see Yalcin I, Bump RC. Validation of two global impression questionnaires for incontinence. Am J Obstet Gynecol. 2003;189(1):98-101). If you decide to extend the study, as suggested above, it would be opportune for you to perform the subsequent follow-ups using PGI-I.
Author Response
Dear reviewers,
Thank you for your careful reading of the manuscript. I really appreciate your helpful comments and suggestions.
Reviewer #1:
1. This is an interesting paper on the influence of vaginal native tissues pelvic floor reconstructive surgery in patients with symptomatic pelvic organ prolapse on preexisting storage Lower Urinary Tract Symptoms.
The study is well-designed with a good sample size. The outcome measures are carefully selected. The subject is useful as it evaluates meshless techniques for vaginal POP surgery, which are expected to become increasingly prevalent.
Its main defect is that it was carried out with an unacceptably short follow-up. Six months is simply not enough time for us to know whether the results of surgery will last. A study of this kind should continue to monitor patients for at least two years. If it is possible for you to extend your study, I would be happy to recommend it for publication at that point.
- Thank you for your comment. We agree that 6 months is a short follow-up period but the aim of our study was not co show the anatomical results of the surgery but the impact of vaginal native tissue repair on pre-existing LUTS and the change in LUTS after operation. Considering the known knowledge regarding impact of medications on LUTS we concluded that 6 weeks is a long enough period of time to know whether the medication works on patient or not and whether the patient feels improvement after treatment. We repeated the examination after 6 weeks and 6 months.
- Secondly, you used PGI-S pre- and post-operatively to monitor the effect of surgery on LUTS, but this is an inappropriate and you should have used PGI-I, which was designed to assess change in the patient’s condition, whereas the PGI-S only assesses severity (see Yalcin I, Bump RC. Validation of two global impression questionnaires for incontinence. Am J Obstet Gynecol. 2003;189(1):98-101). If you decide to extend the study, as suggested above, it would be opportune for you to perform the subsequent follow-ups using PGI-I.
- Thank you for your comment. We used both questionnaires. We added this information to the manuscript. We attach the Table with PGI-I results.
|
|
PGI-I |
||||||
|
|
1very much better |
Much better |
Minimally |
No change |
Mininally worse |
Much worse |
Very much worse |
|
GROUP 1 – URGENCY |
|||||||
|
(T2) 6 weeks after the procedure |
11 (20%) |
26 (47%) |
8 (14,6%) |
5 (9%) |
4 (7,3%) |
1 (1,8%) |
0 |
|
(T3) 6 months after the procedure |
13 (24%) |
23 (42%) |
7 (13%) |
6 (11%) |
5 (9%) |
1 (1,8%) |
0 |
|
GROUP 2 – SUI |
|||||||
|
(T2) 6 weeks after the procedure |
2 (10%) |
6 (30%) |
3 (15%) |
5 (25%) |
3 (15%) |
1 (5%) |
0 |
|
(T3) 6 months after the procedure |
3 (15%) |
6 (30%) |
4 (20%) |
3 (15%) |
3 (15%) |
1 (5%) |
0 |
|
GROUP 3 – MUI |
|||||||
|
(T2) 6 weeks after the procedure |
11 (15,3%) |
24 (33,3%) |
17 (24%) |
12 (17%) |
7 (9,7%) |
1 (1,4%) |
0 |
|
(T3) 6 months after the procedure |
10 (13,8%) |
27 (37,5%) |
15 (21%) |
11 (15,3%) |
8 (11,1%) |
1 (1,4%) |
0 |
|
GROUP 4 - No Clinically significant LUTS SYMPTOMS |
|||||||
|
(T2) 6 weeks after the procedure |
0 |
0 |
6 (11,3%) |
42 (79,2%) |
4 (7,5%) |
1 (1,8%) |
0 |
|
(T3) 6 months after the procedure |
0 |
0 |
6 (11,3%) |
42 (79,2%) |
4 (7,5%) |
1 (1,8%) |
0 |
Reviewer 2 Report
Ewa et al. in their manuscript entitled “The Influence of Vaginal Native Tissues Pelvic Floor Reconstructive Surgery in Patients with Symptomatic Pelvic Organ Prolapse on Preexisting Storage Lower Urinary Tract Symptoms (LUTS)” compared patient reported outcomes in 200 women with symptomatic POP undergoing VNTR on preexisting LUTS. They found that the majority of patients reported improvements in urinary incontinence after surgery, particularly improved urgency.
Major concerns
- The authors note that grade of prolapse affects risk for both pre- and post-operative LUTS. However POP-Q scores are not provided for the study patients. We only know that they have POP-Q scores of two or higher. It is difficult to interpret the results without an indication of the severity of POP.
- The authors demonstrate that the mean results of the surveyed women showed improvement in terms of LUTS scores on the PGI-S, UDI-6, IIQ-7, and ICIQ. While mean scores can be influenced by some women having more drastic improvements than others, it would be helpful to know what percentage of women had improved scores and what percentage did not show improvement or scores that were worse.
- For the women who did not have improved scores is it because they had more severe POP initially or treated at an older age or some other factors?
- The authors state the pelvic health should be defined as lack of POP and urine and stool continence and sexual function (lines 213-214). Was stool continence and sexual function assessed?
Minor concerns
1) The following sentence seems to contradict itself (lines 47-49, page 2): Clinicians should remember that even vaginal tissue repair may be associated with higher rates of recurrence when compared to synthetic mesh repairs, it is outweigh(ed) by (the ) obvious fact that POP surgery with mesh has definitely higher postoperative complication rates.
Author Response
Dear reviewers,
Thank you for your careful reading of the manuscript. I really appreciate your helpful comments and suggestions.
Ewa et al. in their manuscript entitled “The Influence of Vaginal Native Tissues Pelvic Floor Reconstructive Surgery in Patients with Symptomatic Pelvic Organ Prolapse on Preexisting Storage Lower Urinary Tract Symptoms (LUTS)” compared patient reported outcomes in 200 women with symptomatic POP undergoing VNTR on preexisting LUTS. They found that the majority of patients reported improvements in urinary incontinence after surgery, particularly improved urgency.
Major concerns
1.The authors note that grade of prolapse affects risk for both pre- and post-operative LUTS. However POP-Q scores are not provided for the study patients. We only know that they have POP-Q scores of two or higher. It is difficult to interpret the results without an indication of the severity of POP.
- Thank you for your comment. The subsequent changes have been made according to reviewer’s suggestion and the manuscript has been rewritten. We added the sentence:” The POP-Q assessment revealed 58 (29%) women at stage 2, 117 (58.5%) at stage 3 and 25 (12.5%) at stage 4” to the results section of the manuscript.
- The authors demonstrate that the mean results of the surveyed women showed improvement in terms of LUTS scores on the PGI-S, UDI-6, IIQ-7, and ICIQ. While mean scores can be influenced by some women having more drastic improvements than others, it would be helpful to know what percentage of women had improved scores and what percentage did not show improvement or scores that were worse.
- Thank you for your comment. The subsequent changes have been made according to reviewer’s suggestion and the manuscript has been rewritten. We added information regarding improvement to the manuscript.
Table. The relationship between PGI-I and the severity of functional disorders 6 weeks after the surgery
|
Questionnaire |
PGI-I after 6 weeks |
|||||||
|
URGENCY
|
SUI
|
MUI
|
NO symptoms
|
|||||
|
UDI-6 |
0,33 |
p=,015 |
0,31 |
p=,178 |
0,27 |
p=,020 |
0,03 |
p=,821 |
|
IIQ-7 |
0,20 |
p=,152 |
0,43 |
p=,056 |
0,30 |
p=,010 |
0,21 |
p=,134 |
|
ICIQ |
0,22 |
p=,108 |
0,48 |
p=,034 |
0,35 |
p=,002 |
0,29 |
p=,035 |
In the group of women with MUI better PGI-I score 6 weeks after surgery is associated with a lower severity of LUTS, a lower severity of UI and its effect on activity, interpersonal relationships and feelings, and lower intensity of leaking. In women with SUI and those without LUTS , a better PGI-I score 6 weeks after the surgery coincides with lower intensity of leaking, while in the group with urgency with lower intensity of LUTS.
Table. The relationship between PGI-I and the severity of functional disorders 6 months after surgery
|
Questionnaire |
PGI-I after 6 months |
|||||||
|
|
URGENCY
|
SUI
|
MUI
|
NO symptoms
|
||||
|
UDI-6 |
0,3396 |
p=,011 |
0,5165 |
p=,020 |
0,5267 |
p=,000 |
0,1893 |
p=,175 |
|
IIQ-7_ |
0,2979 |
p=,027 |
0,6401 |
p=,002 |
0,4142 |
p=,000 |
0,3425 |
p=,012 |
|
ICIQ |
0,0011 |
p=,994 |
0,6402 |
p=,002 |
0,4488 |
p=,000 |
0,272 |
p=,049 |
In the group of women with MUI and SUI, a better PGI-I score 6 months after surgery is associated with lower severity of LUTS, a lower severity of UI and its effect on activity, relationships and feelings, and lower intensity of leaking. In women with urgency, a better PGI-I score 6 months after surgery is associated with lower severity of LUTS, lower intensity of UI and its effect on activity, interpersonal relationships and feelings, while in the group without UI with lower intensity of UI and its impact on activity, interpersonal relationships and feelings, and less severe leaking.
- For the women who did not have improved scores is it because they had more severe POP initially or treated at an older age or some other factors?
- Thank you for your comment. There was no correlation between age and POP grade.
- The authors state the pelvic health should be defined as lack of POP and urine and stool continence and sexual function (lines 213-214). Was stool continence and sexual function assessed?
-- Thank you for your comment. We assessed the stool continence and sexual function but we will present the results of these in future publication after longer follow-up period.
Minor concerns
1.The following sentence seems to contradict itself (lines 47-49, page 2): Clinicians should remember that even vaginal tissue repair may be associated with higher rates of recurrence when compared to synthetic mesh repairs, it is outweigh(ed) by (the ) obvious fact that POP surgery with mesh has definitely higher postoperative complication rates.
-- Thank you for your comment. The subsequent changes have been made according to reviewer’s suggestion and the manuscript has been rewritten.
Round 2
Reviewer 1 Report
Dear Authors,
this document, in my opinion, is now ready for publication. You had inserted new text and made modifications in answer to my comments and suggestions, and have satisfied all the points I made.